# Cryo-EM structures of human magnesium channel MRS2 reveal gating and regulatory mechanisms

Louis Tung Faat Lai [1], Jayashree Balaraman[1], Fei Zhou[1] & Doreen Matthies [1] ✉

Magnesium ions ($Mg^{2+}$) play an essential role in cellular physiology. In mitochondria, protein and ATP synthesis and various metabolic pathways are directly regulated by $Mg^{2+}$. MRS2, a magnesium channel located in the inner mitochondrial membrane, mediates the influx of $Mg^{2+}$ into the mitochondrial matrix and regulates $Mg^{2+}$ homeostasis. Knockdown of MRS2 in human cells leads to reduced uptake of $Mg^{2+}$ into mitochondria and disruption of the mitochondrial metabolism. Despite the importance of MRS2, the $Mg^{2+}$ translocation and regulation mechanisms of MRS2 are still unclear. Here, using cryo-EM we report the structures of human MRS2 in the presence and absence of $Mg^{2+}$ at 2.8 Å and 3.3 Å, respectively. From the homo-pentameric structures, we identify R332 and M336 as major gating residues, which are then tested using mutagenesis and two cellular divalent ion uptake assays. A network of hydrogen bonds is found connecting the gating residue R332 to the soluble domain, potentially regulating the gate. Two $Mg^{2+}$-binding sites are identified in the MRS2 soluble domain, distinct from the two sites previously reported in CorA, a homolog of MRS2 in prokaryotes. Altogether, this study provides the molecular basis for understanding the $Mg^{2+}$ translocation and regulatory mechanisms of MRS2.

Magnesium ($Mg^{2+}$), the most abundant divalent cation in living organisms, plays an essential role in many biological processes, including ATP synthesis and hydrolysis, DNA replication, protein synthesis, modulation of enzymatic activity, and protein stability[1,2]. It acts as a cofactor of more than 600 enzymes, including protein kinases, ATPases, exonucleases, and other nucleotide-related enzymes. It has been shown that $Mg^{2+}$ is involved in various physiological functions such as muscle contraction, vasodilation, neuronal signaling, and immunity[1-3]. Intracellular $Mg^{2+}$ concentrations are tightly regulated and dysregulation of $Mg^{2+}$ homeostasis is associated with diseases including muscular dysfunction, bone wasting, immunodeficiency, cardiac syndromes, neuronal disorders, obesity, Parkinson's disease, and cancer[2,4–6].

The total $Mg^{2+}$ content in cells amounts to between 17–30 mM, however, most $Mg^{2+}$ is bound to ATP and other molecules, resulting in much lower free $Mg^{2+}$ concentrations between 0.5–1.2 mM[2,7]. In mitochondria, where ATP synthesis and various metabolic processes occur, $Mg^{2+}$ modulates enzymes involved in tricarboxylic acid cycle (TCA), oxidative metabolism and directly participates in mitochondrial metabolism in form of MgATP[2-8,9]. Although the high inner mitochondrial membrane potential (about −180 mV) can theoretically drive a large electrophoretic $Mg^{2+}$ influx into the mitochondrial matrix, the free $Mg^{2+}$ concentration in the matrix is found to be comparable to that in the cytosol (~0.8 mM), implying $Mg^{2+}$ transport is tightly regulated in order to maintain normal mitochondrial physiology[10]. Most of the $Mg^{2+}$ influx into mitochondria is mediated by the Mitochondrial RNA Splicing 2 (MRS2) channel, located in the inner mitochondrial membrane[11-14]. Knockdown of MRS2 in human cells leads to reduced uptake of $Mg^{2+}$ into mitochondria, loss of respiratory complex I, disruption of mitochondrial metabolism, and cell death[15,16]. A loss of

[1]Unit on Structural Biology, Division of Basic and Translational Biophysics, Eunice Kennedy Shriver National Institute of Child Health and Human Development, National Institutes of Health, Bethesda, MD 20892, USA. ✉e-mail: doreen.matthies@nih.gov

function mutation disrupting MRS2 is also associated with demyelination syndrome in rats[17]. Recent studies in mice show that MRS2 is required for lactate-mediated $Mg^{2+}$-uptake in mitochondria[18], and knockout of MRS2 causes reprogramming of the metabolism including upregulation of thermogenesis, oxidative phosphorylation and fatty acid catabolism via HIF1α transcriptional regulation[19]. Despite the physiological implications of MRS2, little is known about its $Mg^{2+}$ translocation and regulatory mechanisms.

MRS2 belongs to the CorA protein superfamily characterized by a highly conserved Glycine-Methionine-Asparagine (GMN) motif in the loop between transmembrane helix 1 and 2. In prokaryotes, CorA regulates the intracellular $Mg^{2+}$ concentration through a negative feedback mechanism, where at low $Mg^{2+}$ concentrations, unbinding of $Mg^{2+}$ ions from the CorA soluble domain favors a series of conductive states, allowing $Mg^{2+}$ permeation[20,21]. MRS2 and CorA only share a sequence identity of ~14%, so insight into MRS2 $Mg^{2+}$ translocation and regulatory mechanism based on CorA has been limited. Until very recently, the only MRS2 structure reported is the monomeric N-terminal soluble domain of yeast MRS2[22], lacking the transmembrane domain. A recent study of the N-terminal domain of human MRS2 revealed it forms a dimer, contrasting the pentameric assembly of CorA[23]. However, the missing structural information of the MRS2 full-length protein including the pore region hampers the understanding of how $Mg^{2+}$ translocation occurs through the pore and how MRS2 is regulated.

Here, we use single particle cryo-EM to determine the structures of human MRS2 in the presence and absence of $Mg^{2+}$. N-terminal sequencing of MRS2 revealed the mitochondrial transit peptide cleavage site at residue 71. From the homo-pentameric structures, M336 and R332 are identified as key gating residues, which are further tested by mutagenesis, and $Mg^{2+}$-dependent growth and $Ni^{2+}$-sensitivity assays. The structures also reveal that MRS2 possesses two $Mg^{2+}$-binding motifs in the soluble domain between neighboring subunits, which are different from known bacterial CorA structures. Finally, an inter-subunit salt bridge between R116 and E291 in the soluble domain is found. Disruption of this salt bridge leads to an increase in channel activity. Altogether, this study provides insight into how translocation of $Mg^{2+}$ is mediated via MRS2 and how it is regulated.

## Results
### Expression, purification, and biochemical characterization of human MRS2
MRS2, a mitochondrial membrane protein encoded in chromosome 6 of the nuclear genome, possesses a mitochondrial transit peptide (MTP) at its N-terminus, which is cleaved after translocation into the inner mitochondrial membrane. As such, we expressed human MRS2 conjugated with a C-terminal FLAG tag in Expi293F cells and purified it in the presence of $Mg^{2+}$ using affinity and size-exclusion chromatography (Supplementary Fig. 1). Purified MRS2 shows a band slightly above 38 kDa based on SDS-PAGE (Supplementary Fig. 1b), which is smaller than the full-length size at 51 kDa, suggesting the N-terminal MTP has been cleaved in purified MRS2. Native-PAGE of purified MRS2 shows a major band between the native marker at 242 kDa and 480 kDa indicating MRS2 assembles into an oligomer (Supplementary Fig. 1c). Negative-staining EM followed by 2D classification of MRS2 particles shows a funnel-shaped structure, resembling that observed in CorA (Supplementary Fig. 1d). To identify the cleavage site of the MTP in human MRS2, online prediction webservers were initially used. The cleavage site was not detected using Target-2.0[24] and SignalP-6.0[25], while it was predicted at residue 21 by Mitofates[26]. These prediction results were contrary to the Uniport entry (Q9HD23) which suggested a cleavage at residue 50. Here, we used an experimental approach, N-terminal sequencing, of purified human MRS2 and the first amino acid detected is residue 71 starting with threonine (Supplementary Fig. 1e). To confirm the identity of the MTP, full-length MRS2 or

truncated MRS2(71-443) conjugated with GFP were expressed in Expi293F cells and imaged using confocal microscopy. It shows that the full-length MRS2-GFP localizes in mitochondria (Supplementary Fig. 1f), while MRS2(71-443)-GFP can no longer be imported into mitochondria and largely localizes in the ER (Supplementary Fig. 1g).

### Structural analysis of human MRS2 in the presence of $Mg^{2+}$ using cryo-EM
Cryo-EM images of purified MRS2 were recorded using a 300 kV Titan Krios equipped with a K3 camera and energy filter (Supplementary Fig. 2a). Initial 2D class averages (Supplementary Fig. 2b) confirmed the by negative-staining EM observed pentameric arrangement of human MRS2, which has been shown for bacterial[21,27–31] and archael[32] members of this protein family and is contrary to the most recent prediction of human MRS2 to form a dimer[23]. The final 3D reconstruction of MRS2 in the presence of $Mg^{2+}$ (referred to as MRS2-$Mg^{2+}$) with C5 symmetry and without symmetry applied (C1) resulted in maps with average resolutions of 2.8 Å and 3.1 Å, respectively, with the highest local resolution estimated to be 2.3 Å (Fig. 1 and Supplementary Figs. 2c–f, 3–4).

The human MRS2 pentamer exhibits a funnel-shaped structure with dimensions of approximately 130 Å by 100 Å by 100 Å (H x W x D) (Fig. 1). No ordered density representing detergent molecule or co-purified lipid is observed in the transmembrane region. The overall architecture of MRS2 resembles that of homologs in the CorA superfamily including *Tm*CorA[21,27–30], *Ec*CorA[31], MjCorA[32], EcZntB[33], and PaZntB[34] (Supplementary Fig. 5), even though they share low sequence identity (Supplementary Fig. 6).

Each of the human MRS2 protomers consists of a large N-terminal soluble domain facing the mitochondrial matrix side. The N-terminal soluble domain is composed of six anti-parallel β-strands and seven α-helices arranged in a α/β/α fold manner (Fig. 1c, d). The arrangement of the α/β/α fold in MRS2 is different from that of *Tm*CorA, which possess seven anti-parallel β-strands and six α-helices (Fig. 1d and Supplementary Fig. 5). The soluble domain connects to the long α8/TM1 helix spanning a total of 71 residues (residues 290–360). The α8/TM1 helix starts from the far matrix side of MRS2 while the C-terminal part of α8 enters the inner mitochondrial membrane, serving as the first transmembrane helix TM1. Following the long α8/TM1 helix, a loop in the intermembrane space containing the highly conserved GMN motif connects to the α9/TM2, which ends at the matrix side. A channel pore connecting intermembrane space and matrix is formed between the intertwined α8/TM1 helices from five subunits. Four $Mg^{2+}$ ions (termed $Mg^{2+}$−1, 2, 3, 4) were identified along the pore and two (termed $Mg^{2+}$−5, 6) were found in between neighboring soluble domains in the MRS2-$Mg^{2+}$ structures (Fig. 1b, c and Supplementary Fig. 4).

### $Mg^{2+}$-translocation pathway and gating mechanism
The five TM1 helices form the wall of the translocation pore, while mostly hydrophilic and charged side chains point to the center (Fig. 2a, b). Extra densities corresponding to four $Mg^{2+}$ ions were found in the center along the pore (Fig. 2a, b and Supplementary Fig. 4). Although these extra densities are located at the center of the C5 symmetry axis and should therefore be interpreted carefully, they are also present in the non-symmetric C1 map and with that are unlikely to be artifacts generated by symmetry refinement (Supplementary Fig. 4b). The upper most $Mg^{2+}$ ($Mg^{2+}$-1), closest to the mitochondrial intermembrane space is coordinated by the highly conserved GMN motif. $Mg^{2+}$-1 interacts with a ring of backbone carbonyl oxygens of G360 and amine group of N362 with a distance of 3.8 Å and 4.3 Å, respectively, indicating that the $Mg^{2+}$-1 is in hydrated form[35,36]. $Mg^{2+}$-2, $Mg^{2+}$-3 and $Mg^{2+}$-4 are coordinated by a ring of hydroxyl oxygens of T346, carbonyl oxygens of N339, and carboxylic oxygens of D329 with a distance of 5.2 Å, 4.3 Å and 3.5 Å, respectively. A highly negatively charged surface was observed in the loop located

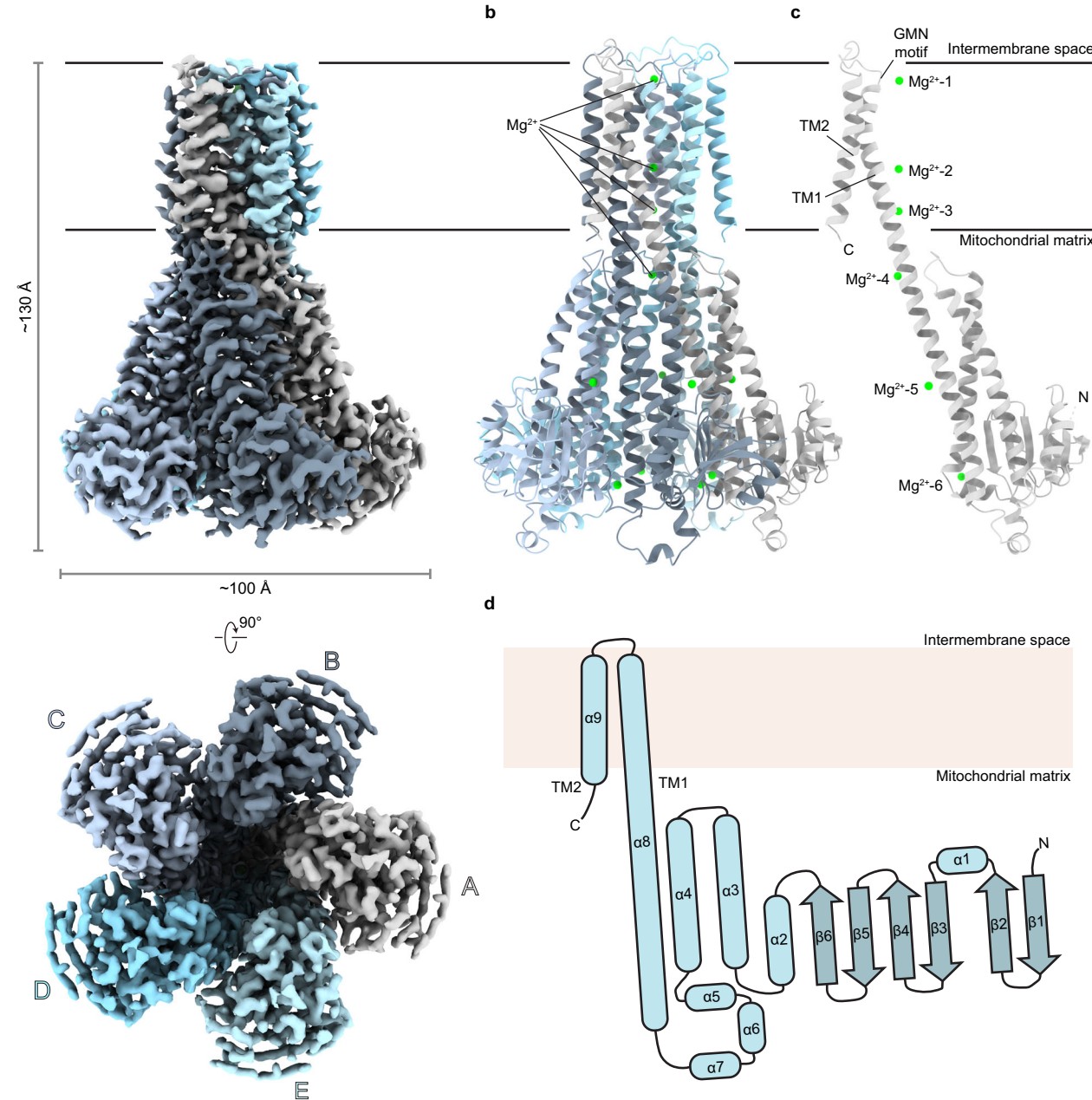

**Fig. 1 | Structure of human MRS2 in the presence of Mg²⁺.** **a** The 2.8 Å cryo-EM density of MRS2 shown in side and bottom views. Density corresponding to each subunit is delineated by different colors. **b, c** Fitted structural model of MRS2 shown as pentameric complex (**b**) and as single subunit (**c**). Magnesium ions bound to MRS2 are represented as green spheres. **d** Topology of MRS2 showing α-helices as rods and β-strands as slightly darker arrows.

at the intermembrane space and the soluble domain in the matrix facing the pore (Fig. 2a). A similar surface has also been observed in CorA and is believed to provide a pulling force for the transport of Mg²⁺ ions[28,30]. The radius plot estimated by the program HOLE[37] shows the narrowest region of the pore is formed by a ring of M336 side chains, with a pore radius of 1.7 Å, followed by R332, and N362/G360, which is part of GMN motif (Fig. 2c). Owing to (1) the hydrophobicity of M336, (2) the repulsive positive charge at R332, and (3) a pore size that is clearly too narrow to allow hydrated Mg²⁺ to pass through, M336 and R332 likely represent candidates to act as gating residues capable of interrupting the ion flow. M336 is conserved among members of the CorA family, for instance, it corresponds to M291 in *Tm*CorA (Supplementary Figs. 5, 6). *Tm*CorA relies on hydrophobic gates M291, L294, M302 (referred to as MM stretch) to control Mg²⁺ flux[38–40]. Interestingly, R332 is conserved among eukaryotes and for

example also found in yeast Mrs2 but not in prokaryotic CorA homologs (Supplementary Fig. 6), suggesting the Mg²⁺-translocation mechanism may be different in eukaryotic MRS2 compared to pro-karyotic CorA. In MRS2, the positive charge of R332 is partially masked by D329 from the neighboring subunit one helix turn below, lowering the energy required for Mg²⁺ to pass through the R332 gate (Fig. 2b, d, e). Furthermore, we found a network of hydrogen bonds from S224 in the loop between soluble α-helix 3–4 of an adjacent subunit to N333 and the potential gating residue R332 (Fig. 2d, e). This network might allosterically link the soluble domain and pore gating residue R332, opening the possibility of a regulation mechanism in which the soluble domain also plays a role.

To evaluate the potential gating role of R332 and M336, we tested Mg²⁺-translocation activity of R332A and M336A mutants by using a Mg²⁺-dependent growth assay as described previously[41,42]. The Mg²⁺-

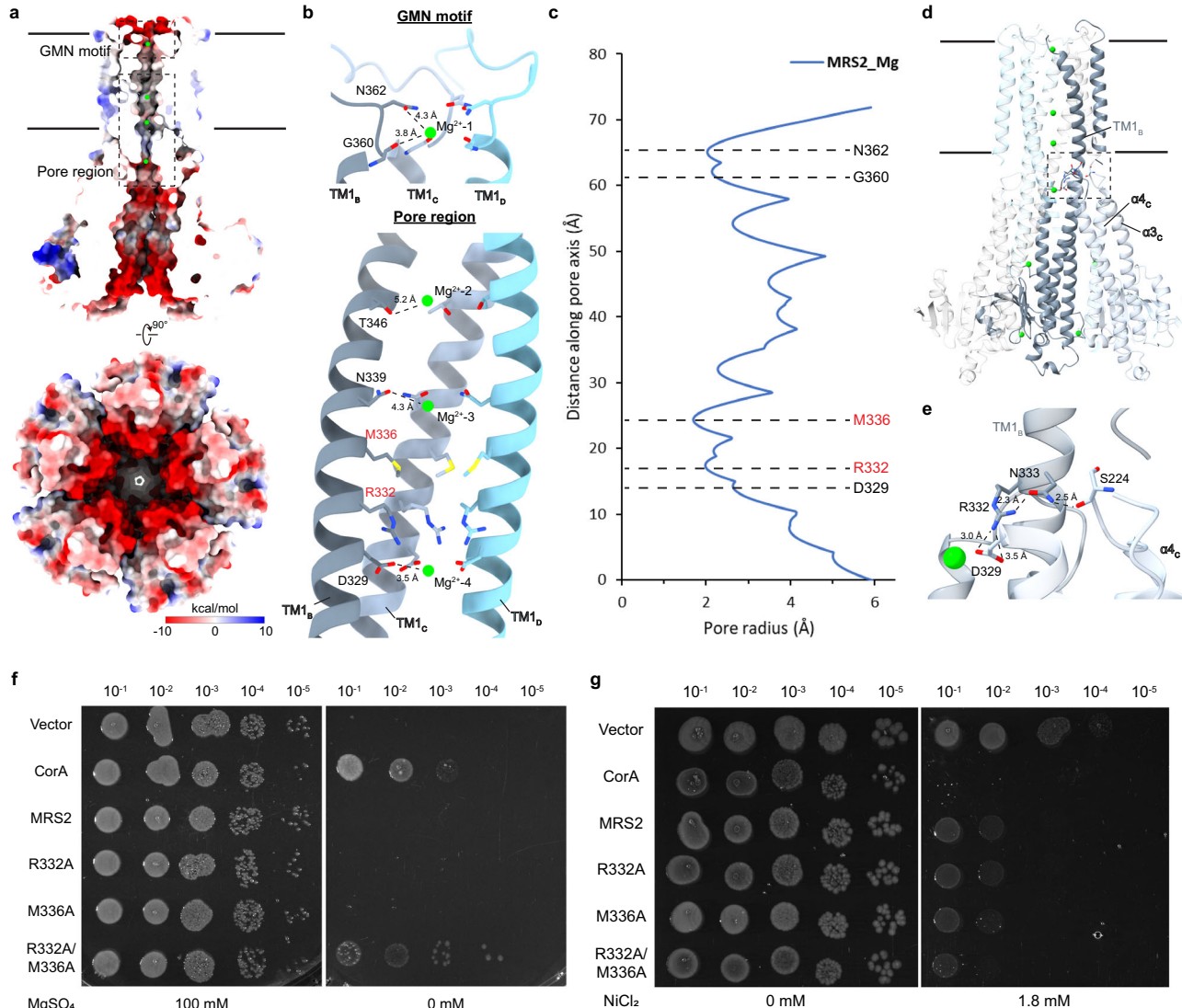

**Fig. 2 | Mg²⁺-translocation pathway in MRS2. a** Sliced side view of MRS2 illustrating the translocation pathway of Mg²⁺. The electrostatic potential of the surface is colored (from negative in red to positive in blue). Insets refer to regions highlighted in (**b**). **b** Enlarged view of the GMN motif and pore region highlighting residues coordinating Mg²⁺ ions. Potential gating residues M336 and R332 are labeled in red. Only helices from subunits B, C, and D are shown for clearer illustration. **c** Pore radius plot of MRS2. **d**, **e** Network of hydrogen bonds connecting gating residue R332 and the loop between soluble α-helix 3–4 of an adjacent subunit. Enlarged view of the interaction is shown in (**e**). **f** Mg²⁺-auxotrophic growth complementation assay using the empty vector and *Tm*CorA (controls), MRS2 and its mutants. Serially diluted Mg²⁺-auxotrophic *E. coli* (BW25113 Δ*mgtA*, Δ*corA*, Δ*yhiD* DE3) containing corresponding plasmids were spotted onto LB plates with and without MgSO₄, and grown at 30 °C. **g** Ni²⁺-sensitivity assay of BL21-(DE3) expressing *Tm*CorA (control), MRS2 and its mutants. Serially diluted *E. coli* containing corresponding plasmids were spotted onto LB plates with and without 1.8 mM NiCl₂, and grown at 30 °C overnight.

auxotrophic *E. coli* strain BW25113 lacking the major Mg²⁺-transporters/channels (*mgtA*, *corA*, *yhiD*) can only grow either by supplementing the growth medium with high Mg²⁺ concentrations, or by complementation with a functional Mg²⁺-transporter/channel. Although expression of the WT *Hs*MRS2 and single gating mutants R332A or M336A is not sufficient to support the growth of the strain without the supplementation of Mg²⁺, double mutant R332A/M336A complements the growth synergistically (Fig. 2f).

A Ni²⁺-sensitivity assay has been employed in characterizing magnesium channel MgtE and its mutants previously[41,43]. As coordination of Ni²⁺ is similar to that of Mg²⁺, and Mrs2 in yeast is able to translocate Ni²⁺ ions like CorA, although it has a higher selectivity for Mg²⁺ over Ni²⁺[14,36], we utilized the toxicity of Ni²⁺ to *E. coli* to evaluate Ni²⁺-uptake and *Hs*MRS2 channel activity. *E. coli* expressing WT *Hs*MRS2 shows a higher Ni²⁺-sensitivity over the negative control of *E. coli* with empty vector only, indicating *Hs*MRS2 does have some

channel activity in *E. coli* (Fig. 2g). Notably, there is no significant increase in Ni²⁺-sensitivity in single gating mutants R332A or M336A. Double gating mutant R332A/M336A shows increased Ni²⁺-sensitivity indicating more uptake of Ni²⁺ compared to WT or single mutants. Together with the complementation growth assay using the Mg²⁺-auxotrophic *E. coli* strain, it suggests both R332 and M336 residues are involved in gating of Mg²⁺ and other divalent cations in *Hs*MRS2, while both gates need to be opened to allow Mg²⁺ to pass through.

## Mg²⁺-binding sites in the soluble domain and their role in channel activity regulation

Two additional Mg²⁺ ions (Mg²⁺-5 and Mg²⁺-6) have been identified in the interface of neighboring subunits between their soluble N-terminal domains termed soluble sites 1 and 2 (Figs. 1b, c and 3a–c). The Mg²⁺ ions in soluble sites 1 and 2 are coordinated largely by negatively charged residues. Mg²⁺-5 and Mg²⁺-6 are hydrated, with some water

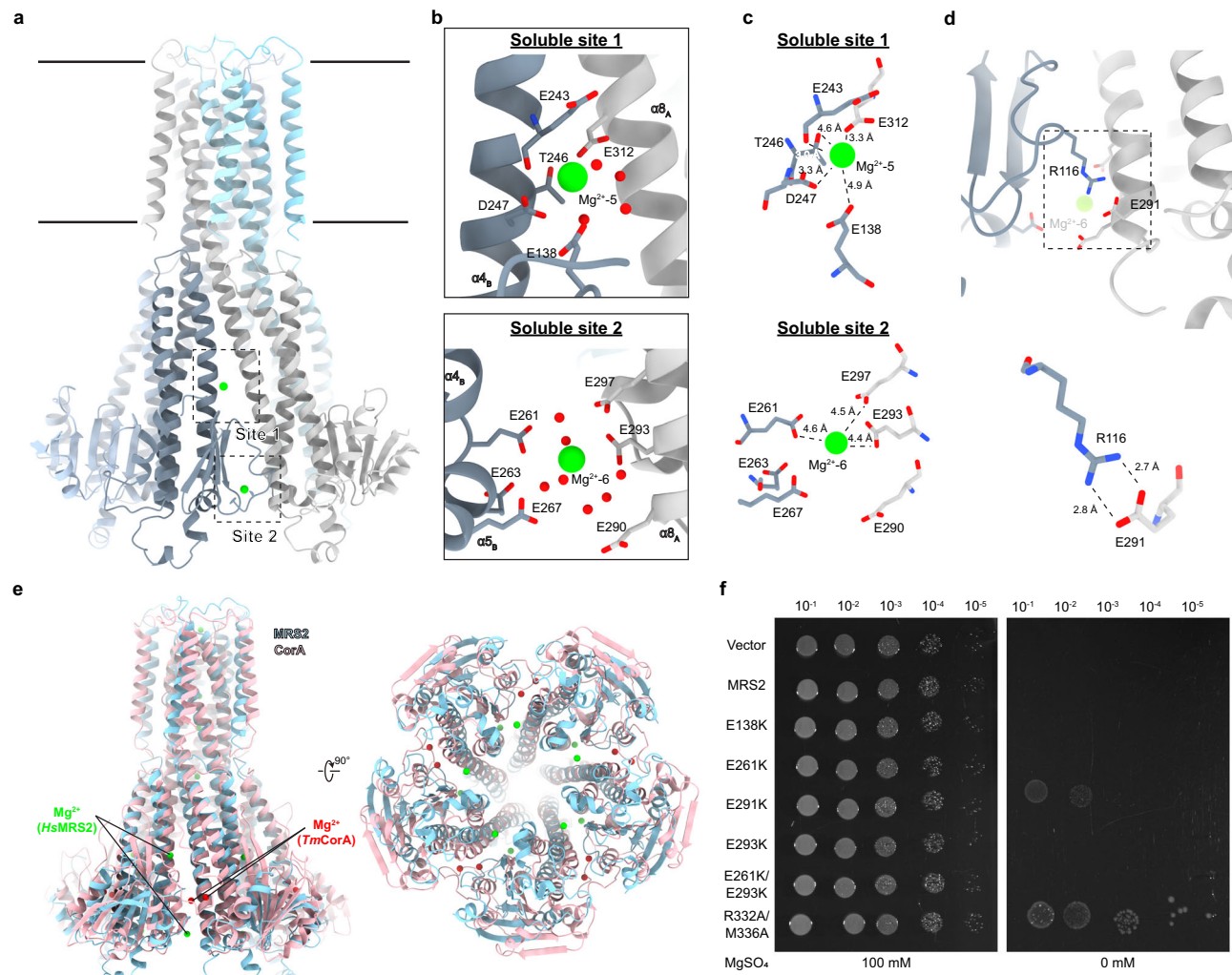

**Fig. 3 | Two Mg²⁺-binding sites in the MRS2 soluble domain and their role in channel activity regulation. a** Two Mg²⁺-binding sites are observed in the MRS2 soluble domain between subunits. Insets refer to regions highlighted in (**b**, **c**). **b** Mg²⁺ in the soluble domain coordinated by negatively charged residues including E243, D247, and E138 from one subunit and E312 from the neighboring subunit in soluble binding site 1. In the soluble binding site 2, Mg²⁺ is coordinated by E261, E263, and E267 from one subunit and E290, E293, E297 from the neighboring subunit. Water molecules in soluble sites 1 and 2 are shown in red spheres. **c** Detailed coordination of Mg²⁺ ions in the soluble site 1 and 2. **d** Salt bridge formed

between R116-E291 pair across subunits. Enlarged view of the R116-E291 is shown (lower panel). **e** Comparison of the Mg²⁺-binding sites in the soluble domain between *Hs*MRS2 and *Tm*CorA. Structure of *Tm*CorA in closed state (PDB: 3JCF) in pink is overlaid with the MRS2 structure in blue. The Mg²⁺ ions in the soluble domain of *Tm*CorA and MRS2 are represented as spheres in red and green, respectively. **f** Mg²⁺-auxotrophic growth complementation assay using MRS2 and its mutants. Serially diluted Mg²⁺-auxotrophic *E. coli* (BW25113 Δ*mgtA*, Δ*corA*, Δ*yhiD* DE3) containing corresponding plasmids were spotted onto LB plates with and without MgSO₄, and grown at 30 °C.

molecules around involved in coordination of Mg²⁺ with distances of 2.1 - 3.1 Å (Fig. 3b). Mg²⁺-5 in site 1 interacts with side chains of T246, D247, E138, carboxyl oxygen of E243, and E312 from the neighboring subunit (Fig. 3b, c). Mg²⁺-6 in site 2 is sandwiched between two negatively charged glutamate clusters from two adjacent subunits. E261, E297 and E293 contribute most in the coordination of Mg²⁺-6 with distances of 4.4–4.6 Å, while E263, E267, and E290 are about 6–7 Å from the Mg²⁺. The Mg²⁺-binding sites in the soluble domain of MRS2 are located at different locations compared to *Tm*CorA and *Ec*CorA structures (Fig. 3e and Supplementary Fig. 5). Sequence alignment also shows the residues involved in soluble site 1 and 2 of MRS2 are not conserved in prokaryotic homologs (Supplementary Fig. 6). When overlaying the *Hs*MRS2 structure to the archaeal *Mj*CorA X-ray structure[32], in which 28 Mg²⁺ ions were modeled into the soluble domain in an asymmetric manner, the Mg²⁺ (atom spec: Mg^E406) between subunit E and A of *Mj*CorA is close to soluble site 1 in MRS2; however, it is coordinated by E155, Y186, T90 and Q62, which is different to the groups of negatively charged residues (E138, E243, D247,

E312) in MRS2 (Supplementary Fig. 5). Interestingly, adjacent to soluble site 2 of *Hs*MRS2, a salt bridge is observed between R116 from one subunit and E291 from the neighboring subunit, with a distance of 2.7–2.8 Å (Fig. 3d, Supplementary Fig. 4c).

It has been proposed that the Mg²⁺-binding sites in the soluble domain of *Tm*CorA serve as a Mg²⁺-sensing motif, where unbinding of Mg²⁺ in those regions under low intracellular Mg²⁺ concentration facilitates channel opening and thereafter Mg²⁺ influx[20,21]. Cryo-EM study of *Tm*CorA shows that it undergoes dramatic asymmetric conformational changes of the soluble domain under low-Mg²⁺ conditions[27]. To investigate whether soluble sites 1 and 2, along with the R116-E291 pair in *Hs*MRS2 are involved in channel regulation, glutamate residues involved in corresponding interactions were mutated to lysine residues and tested using the Mg²⁺-dependent growth assay. Surprisingly, no noticeable change in growth was observed in the soluble Mg²⁺-binding site mutants when compared to WT, while the E291K mutation disrupting the salt bridge between adjacent subunits promotes Mg²⁺-dependent growth, suggesting higher MRS2 channel

activity (Fig. 3f). We speculate that the increase in MRS2 channel activity of the E291K mutant is a result of higher flexibility of the soluble domain that allows the propagation of movement to the TM regions leading to pore opening.

## MRS2 structure under EDTA condition

To investigate whether MRS2 undergoes conformational changes in response to environmental $Mg^{2+}$ concentrations, we depleted most of the free $Mg^{2+}$ in purified MRS2 using dialysis and addition of EDTA and performed structural analysis by cryo-EM. The resulting structures of MRS2 in the presence of 1 mM EDTA (referred to as MRS2-EDTA) with C5 symmetry and without symmetry applied (C1) were determined at an average resolution of 3.3 Å and 3.6 Å, respectively, with the highest local resolution estimated to be 2.7 Å (Fig. 4a, b). Unlike *Tm*CorA, no significant structural changes can be observed when EDTA was included in the sample buffer. The Cα RMSD between MRS2-$Mg^{2+}$ and MRS2-EDTA is 0.481 Å (Supplementary Fig. 7a). While inspecting the $Mg^{2+}$-binding sites found in the *Hs*MRS2 structure in the presence of $Mg^{2+}$, we do observe scattered densities in soluble site 2 but not soluble site 1 (Supplementary Fig. 7b). However, due to (1) the lower overall and local resolution of the MRS2-EDTA structure and (2) under EDTA conditions $Mg^{2+}$ is not expected to be found in the position that is exposed to solvent, we speculate that these extra densities may represent water molecules coordinated by surrounding negatively charged residues. We have not modeled anything into these extra densities due to the high uncertainty. In the central pore regions, densities for $Mg^{2+}$-1, 2, 3 but not for $Mg^{2+}$-4 were observed along the pore of MRS2 even after dialysis and addition of EDTA which should remove most of the free $Mg^{2+}$ in the solution (Fig. 4c). The GMN motif is known to have a high affinity towards $Mg^{2+}$ with an estimated $K_D$ of 1.3 μM[36], which may be able to trap $Mg^{2+}$ in the pore in the presence of EDTA. The conservation of $Mg^{2+}$-1, 2, 3 and the loss of $Mg^{2+}$-4, which is just below the R332/M336-rings, further suggests the gating properties of these residues and that MRS2 remains at low channel open probability without activation.

## Discussion

In this study, we determined the structures of the human mitochondrial magnesium channel MRS2 in the presence of $Mg^{2+}$ and under EDTA condition. MRS2 assembles into a homo-pentamer and displays an overall architecture similar to structures in the CorA family, with slight differences in the secondary structure arrangement (Supplementary Fig. 5). MRS2 possesses a central pore for ion

permeation. Near the intermembrane side of the pore, a conserved GMN motif is located to capture $Mg^{2+}$ ions. Besides the naturally present high membrane potential across the inner mitochondrial membrane, a negative surface potential at the pore-facing side of the soluble domain provides a pulling force for $Mg^{2+}$ influx. Along the pore, R332 and M336 have been identified as the main gating residues. Double mutations of these residues significantly increase $Mg^{2+}$- and $Ni^{2+}$-uptake, suggesting a higher channel activity. Single mutants did not change the channel activity compared to the wild-type. Loss of the $Mg^{2+}$ ion below the M336/R332 gate towards to mitochondrial matrix in the MRS2-EDTA structure further supports the role of M336 and R332 in gating. Strikingly, the R332 gate does not seem to be conserved among prokaryotic CorA homologs, suggesting that MRS2 may adopt a different $Mg^{2+}$-translocation and regulatory mechanism.

In *Tm*CorA, two $Mg^{2+}$-binding sites (M1 and M2) have been identified in the soluble domains of adjacent subunits[28–30]. $Mg^{2+}$ ions are coordinated by D89 and D253 at the M1 site, and D253, E88, L12, D175 at the M2 site. It has been proposed that these M1 and M2 sites serve as divalent cation sensors responding to intracellular ion concentrations and modulating channel activation. Unbinding of $Mg^{2+}$ in the low affinity M1 and M2 sites under low intracellular $Mg^{2+}$ concentration facilitates channel opening. *Tm*CorA D253K mutation abolished the $Mg^{2+}$-dependent protease susceptibility[28] and $Mg^{2+}$-dependent channel inhibition[20]. Previous EPR, cryo-EM and MD-simulation studies of CorA show dramatic movements of the soluble domain under EDTA condition and suggest that the inter-subunit binding of $Mg^{2+}$ in the soluble domain holds the five subunits in the closed, less flexible conformation[20,27,44]. In the present human MRS2-$Mg^{2+}$ structure, two inter-subunit $Mg^{2+}$ ions were also identified but these are in different positions when compared to those in CorA (Fig. 3e and Supplementary Fig. 5), and employ different sets of residues in $Mg^{2+}$ coordination (Supplementary Figs. 5 and 6). Mutating residues in soluble sites 1 and 2 from glutamate (E) to lysine (K) shows no observable differences when compared to WT in the $Mg^{2+}$-dependent growth assay. One possibility is that in MRS2 the E to K mutation facilitates salt bridge formation to the glutamate residues of the adjacent subunit, promoting the closed conformation of the channel. This idea is supported by the loss of function phenotype of the D253K *Tm*CorA mutant using cellular and fluorescence based $Mg^{2+}$-uptake assays[38]. On the other hand, introducing a E291K mutation in *Hs*MRS2, which disrupts a salt bridge between adjacent subunits and likely mimics low-$Mg^{2+}$ conditions, shows a gain of function phenotype (Fig. 3f), potentially due to

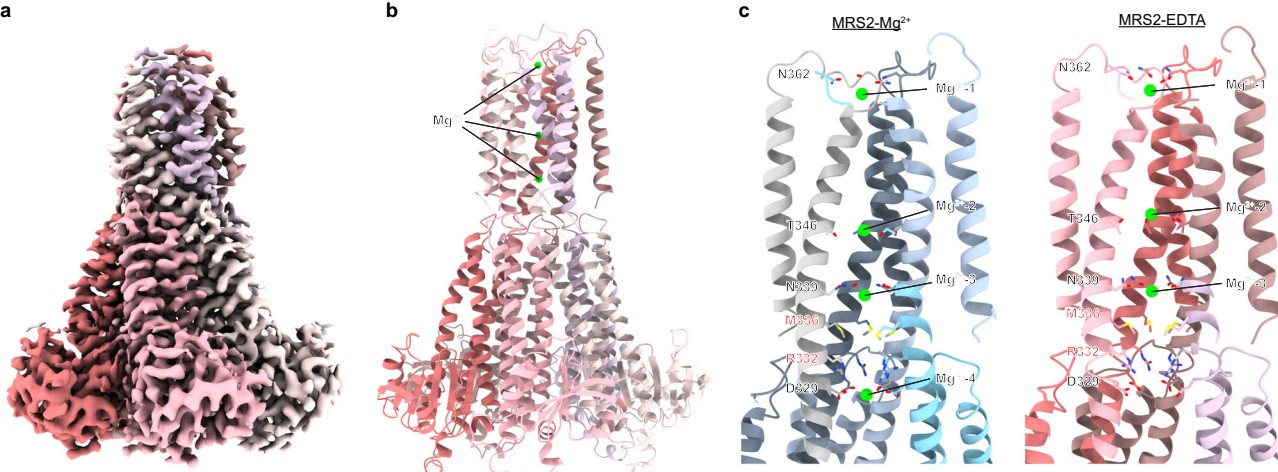

**Fig. 4 | Structure of MRS2 under EDTA condition. a** Cryo-EM density map of MRS2 in the presence of 1 mM EDTA at 3.3 Å. **b** Model of the MRS2-EDTA structure. **c** Comparison of MRS2-$Mg^{2+}$ and MRS2-EDTA showing the loss of $Mg^{2+}$ beneath the R332-ring.

increased flexibility in the soluble domain which might be associated with the regulation of MRS2 channel activity.

During our final manuscript preparation, additional structures of human MRS2 (hMrs2) under various conditions have been reported[45]. As in the present study, hMrs2 assembles into a homo-pentamer and displays a similar structure in the presence of $Mg^{2+}$, low EDTA, high EDTA and without $Mg^{2+}$ and EDTA (termed hMrs2-rest). In general, the reported structures[45] are in agreement with ours, with 0.652 Å Cα RMSD between structures in the presence of $Mg^{2+}$. Two $Mg^{2+}$ ions are identified along the pore of the reported structures, corresponding to $Mg^{2+}$-1, 4 in our study. Two additional $Mg^{2+}$ ions ($Mg^{2+}$-2, 3) have been found in both of our structures, coordinated by a ring of hydroxyl oxygens of T346 and carbonyl oxygens of N339 with a distance of 5.2 Å and 4.3 Å, respectively. We reason that the additional two $Mg^{2+}$ ions identified is likely due to higher $Mg^{2+}$ concentration (40 mM) in our sample, compared to 20 mM used in the previous publication[45]. Interestingly, a tentative $Cl^-$ has been placed in the R332-ring which is proposed to facilitate $Mg^{2+}$-permeation by masking the electrostatic repulsion caused by the R332-ring[45]. Besides, MD-simulation demonstrates that the presence of $Cl^-$ around the R332-ring lowers the energy barrier for $Mg^{2+}$ permeation. When we built the MRS2-$Mg^{2+}$ and MRS2-EDTA models based on our cryo-EM maps, we also see a consistent density above the R332-ring in both structures (Supplementary Fig. 8). Based on the coordination by the R332 residues, we speculate the density may be contributed by a water molecule or possibly an anion such as a chloride ion. Although it is tempting to speculate a chloride ion around the positively charged R332-ring, the presence of a chloride ion in the pore is challenged by the negatively charged pore entry near the GMN motif and the D329-ring, which create a barrier for a chloride ion to enter the pore from either side (Fig. 2a, b). Of note, a previous MD-simulation study on CorA has observed that hydration events occur along the pore including the hydrophobic MM stretch, lowering the free energy barrier for $Mg^{2+}$ permeation[40]. Indeed, water molecules near polar residues T343, N362, E368, and carbonyl oxygens of G356 and V357 along the pore can be found in our MRS2-$Mg^{2+}$ structure. Together, we support an assignment of a water molecule in the density instead of a chloride ion. However, due to the resolution limit in both studies, the identity of the density near R332 remains to be confirmed by additional experimental approaches, such as X-ray anomalous scattering.

Besides a pore dilation and increase in gate flexibility[40], other proposed mechanisms of CorA channel opening include helix rotation of TMs displacing the hydrophobic gating residues[46], as well as relaxation/bending motion of the soluble domains coupled with TM movement leading to gate opening[38,39] (Supplementary Fig. 9). Here, we observed a connection between gating residue R332 and the soluble domain of an adjacent subunit via hydrogen bonds (Fig. 2d, e). We speculate that movement of the soluble domain has a role in the opening of the pore gates in MRS2 (Supplementary Fig. 9e), which is further supported by the increased channel activity upon disrupting the inter-subunit salt bridge between R116 and E291 as mentioned above.

In addition to the gating residue R332, which shows no significant increase in mitochondrial $Mg^{2+}$-uptake when mutated to R332A, R332K, or R332E in the recent study[45], we identified M336 as another gating residue. We found that a single mutation of R332 showed no significant change in channel activity unless M336 was mutated as well, suggesting both gates need to be opened simultaneously or sequentially in order to allow for $Mg^{2+}$ permeation. A previous electrophysiological study shows that matrix $Mg^{2+}$ reduces the open probability of yeast Mrs2, suggesting a negative feedback mechanism[14]. In our study, we observed the loss of $Mg^{2+}$-4 around the D329-ring in the MRS2-EDTA structure (Fig. 4c). It is tempting to propose that the D329-ring plays a role in the negative feedback mechanism of MRS2. Under low matrix $Mg^{2+}$ concentration, $Mg^{2+}$-unbinding from D329 may allow the interaction between D329 and R332, masking the positive charge of R332 and lowering the energy barrier of $Mg^{2+}$ to pass through the R332 gate (Supplementary Fig. 9d). When the $Mg^{2+}$ concentration is high in the mitochondrial matrix, $Mg^{2+}$ is captured by the D329-ring, decoupling the interaction between D329 and R332, re-establishing a barrier by the positive R332-ring. However, the detailed mechanism of how the gating residues are regulated remains to be investigated. Elucidation of the MRS2 structure in an open state will help to address this question.

In summary, our study identifies N-terminal residue 71 as the first amino acid of human MRS2 after protein translocation into the inner mitochondrial membrane and cleavage of its MTP, and reports structures of human MRS2 in the presence and absence of $Mg^{2+}$ at 2.8 Å and 3.3 Å, respectively. We identified R332 and M336 as gating residues and a for stability and channel activity important salt bridge between R116 and E291, which provides further insight into the $Mg^{2+}$ permeation and regulatory mechanism in MRS2.

## Methods

### Protein expression and purification
Full-length human MRS2 (Uniprot Q9HD23) conjugated with a FLAG tag at the C-terminus was expressed in Expi293F cells (Thermo Fisher Scientific, A14527). Cells at a density of $2.0\text{-}3.0 \times 10^6$ cells/ml were transfected with the MRS2-encoding pCMV plasmid using polyethylenimine (PEI) following the protocol described previously[47]. Cells were harvested 48 h after transfection, and resuspended in Lysis buffer (20 mM HEPES, 150 mM NaCl, 10% [v:v] glycerol, 40 mM $MgCl_2$, pH 7.3) supplemented with cOmplete, EDTA-free protease inhibitor cocktail (Roche). After cell disruption by sonication on ice, the membrane fraction was collected by ultracentrifugation at $100,000 \times g$ for 1 h at 4 °C. The pelleted membranes were resuspended with Lysis buffer and pelleted again at $100,000 \times g$ for 1 h. The membrane pellets were flash frozen in liquid nitrogen and stored at −80 °C until use. For solubilization, membranes at a protein concentration of ~4 mg/ml were solubilized with 1% (w:v) n-dodecyl-β-d-maltopyranoside (DDM) (Anatrace) and 0.1% cholesteryl hemisuccinate Tris salt (CHS) (Anatrace) for 2 h at 4 °C, followed by another round of ultracentrifugation to remove insoluble material. The supernatant was incubated with ANTI-FLAG M2 Affinity Gel (Millipore) on an end-over-end rotator for 2 h at 4 °C. The MRS2-bound beads were incubated in Lysis buffer containing 1% lauryl maltose-neopentyl glycol (LMNG) for 30 min at 4 °C, followed by extensive washing with Washing buffer (Lysis buffer supplemented with 0.01% LMNG). Proteins were eluted with 0.2 mg/ml 3× FLAG peptide. Eluted fractions were concentrated to ~3 mg/ml using a 50 kDa cut-off centrifugal filter and loaded onto a Superdex 200 Increase 3.2/300 size-exclusion column (Cytiva) with SEC buffer (20 mM HEPES, 150 mM NaCl, 40 mM $MgCl_2$, 0.003% LMNG, pH 7.3). Peak fractions (~0.5 mg/ml) were collected for cryo-EM grid preparation. For the MRS2 sample in 1 mM EDTA, the sample was prepared in the same procedure except that the sample was dialyzed overnight to remove $MgCl_2$, followed by size-exclusion chromatography with 1 mM EDTA SEC buffer (20 mM HEPES, 150 mM NaCl, 1 mM EDTA, 0.003% LMNG, pH 7.3). The protein fractions were resolved and analyzed using 4–12% Bis-Tris gel (Invitrogen) for SDS-PAGE and 4–16% Bis-Tris gel (Invitrogen) for BN-PAGE.

### N-terminal sequencing
Purified MRS2 was resolved by SDS-PAGE, followed by transferring to methanol activated PVDF membrane in a XCell Mini-Cell Blot Module (Thermo Fisher) at 30 V for 2 h. The MRS2 band on PVDF was stained with Ponceau S, excised and shipped to Creative Proteomics for N-terminal sequencing via Edman degradation.

### Negative staining electron microscopy
3 μL of MRS2 (0.02 mg/mL) was applied on a glow discharged carbon-coated 400 square mesh copper grid (EMS) and incubate for 1 min.

The grid was then blotted by filter paper (Whatman) and washed once with 3 μL of Nano-W negative staining solution (Nanoprobes) followed by incubation with 3 μL of Nano-W for 1 min. The grid was blotted to remove excess staining solution and air-dried by waving. Images were recorded using an FEI Tecnai T20 TEM operated at 200 kV with a direct electron detector K2 Summit (Gatan Inc). Data was collected using SerialEM[48] at a nominal magnification of 25,000x, a pixel size of 3.04 Å/px, and defocus range between −1.5 μm and −2.5 μm. A total of 459 images was collected. Image processing was performed using cisTEM v1.0.0[49]. 215,448 particles were picked from the micrographs after CTF estimation, extracted with a box of 108 px (~328 Å) and analyzed by 2D classification.

## Cryo-EM grid preparation and data collection

3 μl purified MRS2 at approximately 0.5 mg/ml was applied to a glow-discharged 400-mesh R 1.2/1.3 Cu grid (Quantifoil). The cryo grids were blotted for 6 s at 4 °C and 95%, and plunge-frozen into liquid ethane using a Leica EM GP2 (Leica) and stored in liquid nitrogen. The grids were screened on an FEI Tecnai T20 TEM before data collection.

Cryo-EM datasets were acquired with SerialEM[48] using a Titan Krios (FEI, now ThermoFisher Scientific) operated at 300 kV and equipped with an energy filter and K3 camera (Gatan Inc.). Movies of 50 frames with a dose of $1\,e^-/Å^2$ per frame ($50\,e^-/Å^2$ total dose) were recorded at a nominal magnification of 105,000x, corresponding to a physical pixel size of 0.83 Å/px (super-resolution pixel size 0.415 Å/px) in CDS mode at a dose rate of $10\,e^-/px/s$ and a defocus range of −0.7 to −2.0 μm. In total, 3991 and 9656 movies were collected for MRS2-$Mg^{2+}$ and MRS2-EDTA, respectively (Supplementary Table 1).

## Cryo-EM data processing

The overall workflow of image processing is illustrated in Supplementary Fig. 2. All processing was performed within cryoSPARC v3.3.2[50]. Movies were processed with patch motion correction and patch CTF estimation. Good 2D class averages generated from ~1000 manually picked particles served as templates for automatic particle picking.

For MRS2-$Mg^{2+}$, 1,568,444 particles were picked and extracted at 1x binned pixel size of 0.83 Å with a box size of 320 px (~266 Å). Particles were then subjected to 2D classification to remove junk particles. 573,010 good particles selected from one round of 2D classification was subjected to ab-initio reconstruction (K = 2) and heterogenous refinement (K = 2) with C1 symmetry to further sort particles. Non-uniform refinement was performed with 450,554 selected particles, followed by local motion correction and CTF refinement to correct for beam-tilt, spherical aberrations, and per-particle defocus parameters. Non-uniform refinement with polished particles resulted in maps at 2.8 Å (with C5 symmetry applied) and 3.1 Å (with C1 symmetry applied), according to gold-standard FSC = 0.143 criterion (Supplementary Fig. 2).

For MRS2-EDTA, 4,802,706 particles were picked and extracted at 4x binned pixel size of 3.32 Å and sorted by one round of 2D classification, ending up with 1,755,556 particles. Ab-initio reconstruction (K = 3) was performed with the subset of particles after 2D classification, followed by heterogenous refinement (K = 3) using all particles with C1 symmetry was applied to remove junk particles. Particles from one class (2,237,316 particles) were re-extracted at 1x binned pixel size of 0.83 Å and subjected to non-uniform refinement followed by another round of heterogenous refinement (K = 2) to further sort particles. Final non-uniform refinement by using 1,744,117 particles yielded maps at 3.3 Å (with C5 symmetry applied) and 3.6 Å (with C1 symmetry applied) according to gold-standard FSC = 0.143 criterion.

The summary of data collection and imaging processing parameters are shown in Supplementary Tables 1 and 2. The cryo-EM maps

of MRS2·$Mg^{2+}$ (C1 and C5) and MRS2-EDTA (C1 and C5) have been made available and have the following accession codes: EMD−41624, EMD-41629, EMD-41628 and EMD-41630, respectively.

## Model building and refinement

For the atomic model of MRS2-$Mg^{2+}$, predicted MRS2 AlphaFold[51] structure (AF-Q9HD23-F1) was used as an initial model. The initial model was rigid-body fitted into the local resolution filtered map using UCSF Chimera v.1.16[52]. The model was then manually rebuilt in COOT v.0.9.7[53] using the local resolution filtered map, which was generated from refinement with C5 applied. The $Mg^{2+}$ ions assigned in the pore regions were confirmed in the C1 map. Loop regions (residues 174−181, 273−287) were built with the unsharpened map. Iterative rounds of manual refinement in COOT and real-space refinement in Phenix v.1.20.1-4487[54] were performed. For the MRS2-EDTA structures, models were generated from multiple rounds of manual refinement and real-space refinements with the final model of MRS2-$Mg^{2+}$ as initial model. The quality of the model and fit to the density was assessed using MolProbity[55] and Phenix[54]. All structural figures were prepared using UCSF Chimera v.1.16[52] or UCSF ChimeraX v.1.4[56]. PDBs have been made available together with the EM maps with the following PDB IDs: 8TUL (MRS2-$Mg^{2+}$) and 8TUP (MRS2-EDTA).

## Subcellular localization of MRS2

Expi293F cells at a density of $2.0\times10^6$ cells/ml were transfected with the MRS2-GFP or MRS2(71-443)-GFP encoding pCMV plasmid using polyethylenimine (PEI) following the protocol described previously[47]. 24 h after transfection, the cells were stained with 100 nM MitoTracker Red (Thermo Fisher Scientific) or 1 μM ER-Tracker Red (Thermo Fisher Scientific) for 30 min at 37°C. The cells were washed twice with PBS and imaged using a Zeiss LSM 880 confocal laser scanning microscope.

## $Mg^{2+}$-dependent *E. coli* growth assay

The $Mg^{2+}$-auxotrophic *E. coli* strain (BW25113 Δ*mgtA*, Δ*corA*, Δ*yhiD* DE3), which has been used previously[41,42], was transformed with His-*Tm*CorA, MRS2(71-443)-FLAG, its mutants or without insert in a pET28a backbone. The transformants were inoculated and grown in kanamycin (50 μg/mL) containing LB medium (LBK) supplemented with 100 mM MgSO4, shaking at 250 rpm and 37 °C overnight. The overnight cultures were used to inoculate fresh LBK in a 1:100 ratio and grown until they reached an OD600 of 0.6. The bacterial cultures were then diluted 10-fold serially with LBK medium, spotted onto LBK agar plates containing 0.1 mM IPTG and 0 or 100 mM MgSO4, and incubated at 30 °C for 2 days. Plates were imaged using ChemiDoc MP Imaging System (Bio-rad).

## $Ni^{2+}$-sensitivity assay

BL21(DE3) (Sigma, CMC0014) was transformed with His-*Tm*CorA, MRS2(71-443)-FLAG, its mutants or without insert in a pET28a backbone. The transformants were inoculated and grown in LBK medium, shaking at 250 rpm and 37 °C overnight. The overnight cultures were used to inoculate fresh LBK in a 1:100 ratio and grown until they reached an OD600 of 0.6. The bacterial cultures were then diluted 10-fold serially with LBK medium, spotted onto LBK agar plates containing 0.1 mM IPTG, and 0 or 1.8 mM NiCl2, and incubated at 30 °C overnight. Plates were imaged using ChemiDoc MP Imaging System (Bio-rad).

## Reporting summary

Further information on research design is available in the Nature Portfolio Reporting Summary linked to this article.

## Data availability

The data that support this study are available from the corresponding authors upon request. Cryo-EM maps have been deposited in the

Electron Microscopy Data Bank (EMDB) under accession codes EMD-41624 (Cryo-EM structure of the human MRS2 magnesium channel under $Mg^{2+}$ condition); EMD-41628 (Cryo-EM structure of the human MRS2 magnesium channel under $Mg^{2+}$-free condition); EMD-41629 (Cryo-EM structure of the human MRS2 magnesium channel under $Mg^{2+}$ condition (C1 map)); and EMD-41630 (Cryo-EM structure of the human MRS2 magnesium channel under $Mg^{2+}$-free condition (C1 map)). The atomic coordinates have been deposited in the Protein Data Bank (PDB) under accession codes 8TUL (Cryo-EM structure of the human MRS2 magnesium channel under $Mg^{2+}$ condition); and 8TUP (Cryo-EM structure of the human MRS2 magnesium channel under $Mg^{2+}$-free condition). The source data for Supplementary Fig. 1b, c are provided in the Source Data file. Source data are provided with this paper.

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

## Acknowledgements
We thank Joshua Zimmerberg, Jennifer Petersen, and Paul Blank for access to a Tecnai T20 electron microscope and Zeiss LSM 880 confocal microscope; Allison Zeher and Rick K. Huang for support on the Krios; and Stéphane Mahé and Joe Cometa for technical support on the T20 and Krios. We are grateful to Prof. Dr. Koichi Ito for providing us the Mg$^{2+}$-auxotrophic *E. coli* strain (BW25113 Δ*mgtA*, Δ*corA*, Δ*yhiD* DE3). We also like to thank Eduardo Perozo and Lesley Earl for helpful comments on the manuscript. This research was supported by the Division of Intramural Research of the *Eunice Kennedy Shriver* National Institute of Child Health and Human Development, NIH (grant NICHD intramural project ZIA HD008998). This work utilized the computational resources of the NIH HPC Biowulf cluster (http://hpc.nih.gov).

## Author contributions
L.T.F.L. performed protein expression, purification, negative-staining EM, cryo-EM screening, cryo-EM data collection, image processing, and the Mg$^{2+}$-dependent growth assay. J.B. performed the Ni$^{2+}$-sensitivity assay and the Mg$^{2+}$-dependent growth assay. F.Z. helped with cryo-EM data collection. L.T.F.L. and D.M. designed the study, performed model building, structural analysis, made figures and wrote the manuscript. All authors discussed the results and contributed to the manuscript preparation.

## Funding

## Competing interests
The authors declare no competing interests.
