## [Peer Review File · Nature Communications]

Cryo-EM structures of human magnesium channel MRS2 reveal gating and regulatory mechanismsREVIEWER COMMENTS

Reviewer #1 (Remarks to the Author):

In this contribution by Lai et al., the Cryo-EM structures of human Mg²⁺-channel Mrs2 are presented with some limited functional insight into the mechanism of transport. The manuscript brings very little novelty as the majority of findings have been already recently reported by the study from Shen group (<https://doi.org/10.1038/s41467-023-40516-2>). Both studies are in good agreement, with some extra insights from the current work - like the MTP cleavage position and the enhanced effect of the double mutation (R332 with M336) on transport, however such details are of interest to a few readers working in the same field.

The manuscript should be in principle rewritten to focus more on comparison of both studies, however again it would be interesting only to limited audience.

Some other concerns:

I fail to grasp the results of Mg²⁺ auxotrophic growth complementation experiment. In my understanding, addition of Mrs2 should phenotypically look similar to CorA as both serve to import Mg²⁺ at low magnesium condition. Here it seems that only when presumable gate is removed - the channel is capable to transport the substrate -which is at least odd. Perhaps solving a structure of this mutant can yield some interesting insights. This experiment also contradicts the following experiment on Ni²⁺-toxicity assay, where Mrs2 behaves similarly to CorA as expected. Furthermore, I do not agree with the authors that the double mutant R332A/M336A shows increased sensitivity to Ni²⁺ - perhaps to some small extent. But the fact that two experiments do not agree with each other is worrisome .

Reviewer #2 (Remarks to the Author):

Lai et al. report two cryoEM structures of the human MRS2 pentamer, one in the presence of Mg²⁺ and one in its absence. Their study is scientifically and technically sound, utilizing state-of-the-art cryoEM analysis. The manuscript is well-written with high-quality illustrations, providing a comprehensive interpretation of the structures. The authors wisely avoid over-interpretation, and their major findings are supported by mutagenesis studies and functional assays.

While structures of bacterial homologues are already available, the mechanistic insights into the gating mechanism of human MRS2 hold significant importance and high interest in the field. It was previously even unclear whether MRS2 forms pentamers.

However, it's worth noting that a similar study by Li et al. (2023) was recently published in the same journal, with slightly higher resolutions and more experimental conditions, including higher EDTA concentrations and the absence of external Mg²⁺. Li et al. identified the R-ring along the channel's pore, which they suggested that it may function as a charge repulsion barrier. In close proximity they identified a Cl⁻. Based on MD simulations, they suggested that it may function as a ferry to jointly gate Mg²⁺ permeation, an overall intriguing mechanism.

In the structures reported here, the authors confirm the presence of the R-ring as a crucial pore component. They also observe a density in the expected position of Cl⁻, similar to Li et al., but refrain from confirming its presence due to limited resolution.

One notable finding in this study is the confirmation of M336 as another critical gating residue. Interestingly, only a double mutant of the R-ring and M336 shows significant changes in channel activity, suggesting that both gates may need to be opened simultaneously or sequentially, a significant finding.

Additionally, the authors demonstrate that the movement of the soluble domain might play a role in opening the pore gate, as increased channel activity is observed upon disrupting an inter-subunit salt bridge.

However, considering the recently available structures of MRS2, the structural insights provided here are somewhat limited. The model proposed by Li et al. is neither explicitly challenged nor confirmed by this study. The authors should expand their discussion in this direction. A structure in the "open"

state is still lacking, and, as the authors acknowledge, the overall gating mechanism remains unclear. Some hints are provided regarding the regulation of the R-ring by its interaction with D329, but the model in supplementary Figure 9 lacks further experimental support. Molecular dynamics (MD) simulations could be considered to explore this further.

Additional comments:

- The authors should report whether they observed densities for DDM or co-purified lipids.
- The presence of blurry densities below the pore in the 2D classes may suggest the existence of artificial tail-to-tail dimers due to solubilization through hydrophobic interactions. The authors should address this issue and consider reconstitution in nanodiscs to stabilize the channel state.
- The authors should provide an explanation or comment on the higher molecular weight bands observed in BN-PAGE.
- The identification of two additional Mg²⁺ ions along the pore is surprising and should be discussed in more detail.

Reviewer #3 (Remarks to the Author):

The manuscript "Cryo-EM structures of human magnesium channel MRS2 reveal gating and regulatory mechanisms" by Lai and coworkers describes substantial progress in our understanding of the structural basis and mechanism of the human mitochondrial Mg²⁺ transporter.

The paper presents two high-resolution cryo-EM structures of the transporter obtained under Mg-saturating conditions (inactive form) and in the presence of EDTA, which removes Mg²⁺ from regulatory sites but does not empty the pore entirely. This indicates high-affinity pore-ion interactions and also suggests that the ions may play a structure-stabilizing role. The resolution allowed the authors to conclude which Mg²⁺ ion is directly coordinated by the sidechain or backbone oxygens and which one is coordinated through water (Mg²⁺-1, for instance). The identification of R332 and M336 (main constriction) as two partially redundant gates (both have to be mutated to produce a gain-of-function effect) is a novel and interesting result. The identification and disruption of the R116-E291 salt bridge between adjacent soluble domains, which results in a higher transporter activity, are also important results showing a regulatory mechanism different from bacterial homologs. The discovery of the 71-residue-long mitochondrial transit peptide at the N-terminus of MRS2 is novel too.

What makes this work distinct from the recently published paper by Li and coworkers (Ref 45)? The authors of that previous paper found only two Mg²⁺ ions in the pore, and they proposed the role of Cl⁻ ions as co-ions that help Mg²⁺ overcome the ring of R332 residues. They performed MD simulations illustrating the degree of pore hydration (with the accuracy of the force field) and emphasized the role of membrane potential in driving Mg²⁺ inside. But the present paper better shows the effects of mutations, which were tested in-vivo, and the toxicity of Ni²⁺, which also permeates through the transporter, and the new R116-E291 salt bridge with its regulatory role. In the two papers, the structures are essentially identical, which gives credit to both groups and shows how reliably the same structure was determined in different detergents. In other aspects, the two papers perfectly complement each other and both deserve attention.

About Mg²⁺-free condition (1 mM EDTA), under which three Mg²⁺ ions remain in the pore. For Mg²⁺ - EDTA in HEPES, K_d is about 1 μM (O'Brien, J. Chem. Educ. 2015, 92, 1547–1551), which is indeed comparable to the GMN-Mg²⁺ affinity. I would call these conditions not Mg-free, but rather low-Mg condition. That is why the ion remains in the site. If you try a higher EDTA concentration or a stronger chelator, in the absence of Mg²⁺, the GMN motifs and other domains may get disordered. The fact that GMN binds Mg²⁺ so tightly may have a clear functional consequence, namely, a very slow Mg²⁺ transport rate. Because the GMN site is strictly 'in-series' with the rest of the pore pathway, the Mg²⁺ unbinding kinetics might be limiting unless GMN loops come apart and release the ion in the active state. The result begs for experimental measurements of Mg²⁺ unbinding rate. This tight binding is in

stark contrast to K⁺ channels where the ion entering the selectivity filter from water experiences zero free energy change (Varma and Rempe, BJ 2007, 93:1093).

The work is implemented according to high standards with a lot of details in the supplement, so I have no technical comments.

I have only one request for a short additional experiment. The mitochondrial MRS2 protein was expressed in Expi293F cells (apparently derived from the human HEK293 cell line). My question is whether the expressed protein is localized strictly in mitochondria or elsewhere. The standard immunofluorescence with anti-Flag antibodies may quickly give an answer. Also, is overexpression of the R332A/M336A mutant toxic for the cells?

Minor points and suggested edits:

Line 86: 'challenging' can be replaced by 'limited'

Line 112: I would replace 'using SDS-PAGE' with 'based on SDS-PAGE'

Line 169: 'carboxylic acids' better to replace with 'carboxylates' or 'carboxylic oxygens'

Line 178: 'capable of interrupting the ion flow' instead of 'regulating Mg²⁺ translocation'

Lines 216-223: This paragraph describes a very important difference in ion coordination, either directly by sidechain or backbone oxygens or through intercalating water. I would suggest putting a special figure illustrating these types of coordination and two or three phrases in the Discussion. Coordination through water is characteristic specifically for Mg²⁺ and might be one of the components of the selectivity mechanism.

Line 278: I would add 'Besides naturally present high membrane potential across the inner mitochondrial membrane, a negative surface potential at the pore-facing side...'

Line 306: I would add: '... which disrupts a salt bridge between adjacent subunits and mimics low-Mg²⁺ conditions, shows a gain of function...'

One question about Fig. 3d. It needs to be better discussed whether Mg²⁺⁻⁶ when present in the second regulatory site stabilizes or disrupts the R116-E291 salt bridge. With the E291K mutation, the channel becomes more active, and therefore Mg²⁺⁻⁶ may stabilize this bridge.

Please increase the fonts in the Supplemental figures, some panels are not legible.

Response to reviewers

Reviewer #1 (Remarks to the Author):

In this contribution by Lai et al., the Cryo-EM structures of human Mg²⁺-channel Mrs2 are presented with some limited functional insight into the mechanism of transport.

1. The manuscript brings very little novelty as the majority of findings have been already recently reported by the study from Shen group (<https://doi.org/10.1038/s41467-023-40516-2>). Both studies are in good agreement, with some extra insights from the current work - like the MTP cleavage position and the enhanced effect of the double mutation (R332 with M336) on transport, however such details are of interest to a few readers working in the same field. The manuscript should be in principle rewritten to focus more on comparison of both studies, however again it would be interesting only to limited audience.

Response: During the manuscript preparation, we noticed that an independent work studying structures of human MRS2 (hMrs2) under various conditions have been reported by Li and coworkers. We thoroughly reviewed this recent publication and pinpointed that our work provides extra insights into the gating and regulatory mechanisms of the human MRS2 channel.

The paper published by Li et al. has identified R332 as a potential gating residue based on the structural information, however, their attempt on confirming the role of R332 experimentally is not successful, as mutation R332 to A/K/E fails to show significant alteration in Mg²⁺-uptake activity. In our study, we have identified M336 as another gating residue in addition to R332. Mutations of both M336 and R332, but not a single mutation, shows a significant increase in MRS2 activity in two ion uptake assays we performed, confirming the importance of M336 in gating and the need for both gates to be opened to allow Mg²⁺ to pass through the channel. Besides, we have found an inter-subunit salt bridge between R116-E291 involved in channel regulation. Disruption of the R116-E291 salt bridge results in a higher channel activity. Identification of the mitochondrial transit peptide of human MRS2 in our study also facilitates further characterization of this protein in different systems, such as heterologous expression of MRS2 in Xenopus oocytes for electrophysiological measurement on the plasma membrane. Hence, we do not agree with the reviewer's view that this manuscript provides little interest to the field.

In addition, our work is largely in agreement with Li et al.. Our structures are very similar compared to these reported by Li et al., with only 0.652 Å Ca RMSD between structures in the presence of Mg²⁺. The independency of our work also provides validation to each other, which consolidates the confidence and robustness of both works. The reproducibility of scientific research is best revealed when independent investigations of the same problem arrive at similar conclusions (<https://www.nature.com/articles/s41467-020-17817-x>).

Some other concerns:

2. I fail to grasp the results of Mg²⁺ auxotrophic growth complementation experiment. In my understanding, addition of Mrs2 should phenotypically look similar to CorA as both serve to import Mg²⁺ at low magnesium condition. Here it seems that only when presumable gate is removed - the channel is capable to transport the substrate -which is at least odd. Perhaps solving a structure of this mutant can yield some interesting insights.

Response: *Based on both our study and the study by Li et al., MRS2 is different from CorA in terms of gating and regulatory mechanisms. 1) The gating of MRS2 is mediated via M336-ring and R332-ring, while the tight arginine ring is not observed in CorA, implying MRS2 employs a distinct gating mechanism. 2) It has been shown that Mg²⁺ ions in the soluble domain of CorA play a role in regulating channel activity (Dalmas et al., Nat. Comms. 2014, 5(1), 3590). Unbinding of Mg²⁺ triggers dramatic structural changes in the soluble domain of CorA, which promotes channel opening (Matthies et al., Cell 2016, 164(4), 747-756). Surprisingly, no significant conformational changes of MRS2 are observed under low Mg²⁺ environment in both our and the recently published study. It suggests MRS2 is tightly regulated via a mechanism different from that of CorA.*

In the Mg²⁺ auxotrophic growth complementation experiment, WT MRS2 is not sufficient to complement growth of the auxotrophic E. coli strain, unless R332A/M336A mutations were introduced (Fig. 2f). It suggests that either WT MRS2 is not functional in E. coli or it exhibits low activity that is insufficient to complement the growth of the auxotrophic strain. The Ni²⁺-sensitivity assay which showed higher sensitivity when cells are expressing WT MRS2 compared to the empty vector control (Fig.2g), indicates MRS2 does have some activity in E. coli. We introduced additional text to the results in line 210 to better communicate our findings.

Low activity of MRS2 compared to CorA is also observed in the recent study (He et al., *BioRxiv* 2023, <https://doi.org/10.1101/2023.08.12.553106>), where no Mg^{2+} current is detected in MRS2-expressing *Xenopus* oocytes, unless R332S mutation is introduced. The reason of why MRS2 exhibits lower activities remains to be elucidated, which will be part of our future study.

Regarding the suggestion of solving the structures of the R332A/M336A mutant, we anticipate the mutation of these gating residues would only disrupt the gate locally instead of causing significant structural changes. Due to a variety of factors, the structural determination by cryo-EM will take us several months to complete, therefore we used Alphafold 2-multimer to predict the mutant structure (Fig. R1). The predicted structure of the R332A/M336A mutant is almost identical to the predicted WT structure, with only 0.416 Å C α RMSD between ordered region of the structure (residue 84-400).

Fig. R1. Structural analysis of the MRS2 R332A/M336A mutant predicted using Alphafold 2-multimer. **a**, Structures of WT MRS2 and the R332A/M336A mutant predicted using Alphafold 2-multimer are analyzed and compared. The predicted structural model of the R332A/M336A mutant is very similar to the predicted WT model, with only 0.416 Å C α RMSD between the ordered region (residue 84-400). The structural model of the R332A/M336A mutant with one subunit colored according to RMSD between the two predicted structures is shown. **b**, Comparison between the WT and the R332A/M336A mutant in the pore region. Residues facing the translocation pathway

are shown. Other than disruption of the gate locally, no significant conformational change is predicted in the R332A/M336A mutant.

3. This experiment also contradicts the following experiment on Ni²⁺-toxicity assay, where Mrs2 behaves similarly to CorA as expected. Furthermore, I do not agree with the authors that the double mutant R332A/M336A shows increased sensitivity to Ni²⁺ - perhaps to some small extent. But the fact that two experiments do not agree with each other is worrisome.

Response: The Ni²⁺-sensitivity assay we used in our study is an established method characterizing channel activity of divalent cation transport (Jin et al., Plos Biol. 2021, 19(4), e3001231; Tomita et al., Nat. Comms, 2017, 8(1), 148). Owing to the toxicity of Ni²⁺ towards E. coli, higher Ni²⁺-sensitivity (reduction in growth) indicates higher channel activity in translocating divalent cations. Our results show E. coli expressing WT HsMRS2 exhibits a higher Ni²⁺-sensitivity over the negative control of E. coli with empty vector only (Fig. 2g), while double gating mutant R332A/M336A shows increased Ni²⁺-sensitivity, indicating higher channel activity compared to WT or single mutants. The result indicates both R332 and M336 serve as gating residues, and it indeed supports the result of the Mg²⁺ auxotrophic growth assay, instead of contradicting each other.

Reviewer #2 (Remarks to the Author):

Lai et al. report two cryoEM structures of the human MRS2 pentamer, one in the presence of Mg²⁺ and one in its absence. Their study is scientifically and technically sound, utilizing state-of-the-art cryoEM analysis. The manuscript is well-written with high-quality illustrations, providing a comprehensive interpretation of the structures. The authors wisely avoid over-interpretation, and their major findings are supported by mutagenesis studies and functional assays.

While structures of bacterial homologues are already available, the mechanistic insights into the gating mechanism of human MRS2 hold significant importance and high interest in the field. It was previously even unclear whether MRS2 forms pentamers.

However, it's worth noting that a similar study by Li et al. (2023) was recently published in the same journal, with slightly higher resolutions and more experimental conditions, including higher EDTA concentrations and the absence of external Mg²⁺. Li et al. identified the R-ring along the channel's pore, which they suggested that it may function as a charge repulsion barrier. In close proximity they identified a Cl⁻. Based on MD simulations, they suggested that it may function as a ferry to jointly gate Mg²⁺ permeation, an overall intriguing mechanism.

In the structures reported here, the authors confirm the presence of the R-ring as a crucial pore component. They also observe a density in the expected position of Cl⁻, similar to Li et al., but refrain from confirming its presence due to limited resolution.

One notable finding in this study is the confirmation of M336 as another critical gating residue. Interestingly, only a double mutant of the R-ring and M336 shows significant changes in channel activity, suggesting that both gates may need to be opened simultaneously or sequentially, a significant finding.

Additionally, the authors demonstrate that the movement of the soluble domain might play a role in opening the pore gate, as increased channel activity is observed upon disrupting an inter-subunit salt bridge.

1. However, considering the recently available structures of MRS2, the structural insights provided here are somewhat limited. The model proposed by Li et al. is neither explicitly challenged nor confirmed by this study. The authors should expand their discussion in this direction.

***Response:** We thank the reviewer for the thorough review of our manuscript. As mentioned by the reviewer, we observe a density around the R332-ring, consistent with the structure reported in Li et al.. We speculate the density may be contributed by a water molecule or possibly an anion such as a chloride ion. Although it is tempting to speculate a chloride ion around the positively charged R332-ring, the presence of a chloride ion in the pore is challenged by the negatively charged pore entry near the GMN motif and the D329-ring, which create a barrier for a chloride ion to enter the pore from either side (Fig. 2a-b). The MD-simulation in Li et al. shows that Mrs2 with Cl⁻ result in a lower energy barrier for permeation. However, Mrs2 with Cl⁻ initially placed around the R332-ring is used as input model of the MD-simulation, which does not provide information about the likelihood and how Cl⁻ enters the pore and reaches the R332-ring. Given the previous MD-simulation study on CorA which suggests hydration events occur along the pore, lowering the free energy barrier for Mg²⁺ permeation (Neale et al., Plos. Comp. Biol. 2015, 11(7), e1004303), together with the water molecules near polar residues T343, N362, E368, and carbonyl oxygens of G356 and V357 along the pore found in our MRS2-Mg²⁺ structure, we support an assignment of a water molecule in the density instead of a chloride ion. However, we cannot confirm the identity of the density due to the current resolution limit. The identity of the density can be tested by additional experimental approaches, such as X-ray anomalous scattering.*

As suggested by the reviewer, we revised and expanded the discussion about the model proposed by Li et al. in line 332-347 highlighted in yellow.

2. A structure in the "open" state is still lacking, and, as the authors acknowledge, the overall gating mechanism remains unclear. Some hints are provided regarding the regulation of the R-ring by its interaction with D329, but the model in supplementary Figure 9 lacks further experimental support. Molecular dynamics (MD) simulations could be considered to explore this further.

Response: *From our structures, we found that D329 is close to the gating residue R332 with a distance of 3.0 - 3.5 Å (Fig. 2d-e), masking the positive charge of R332 and lowering the energy barrier required for Mg²⁺ to pass through the R332 gate. The loss of Mg²⁺ around the D329-ring in the low Mg²⁺ environment (Fig.4c) may allow the movement of D329, which likely couples with the conformational change of R332 via their interaction, providing one route of gate opening. Although we do not have experimental support besides our structural data, it is still tempting to propose this mechanism together with many other proposals to the readers and make it open for discussion and validation. We thank the reviewer for the suggestion to perform MD simulations to test this model. We have already initiated a collaboration with an MD-simulation expert of membrane proteins and MD-simulations are ongoing but will take months to complete and analyze. We are looking forward to sharing our future results in a separate manuscript.*

Additional comments:

3. - The authors should report whether they observed densities for DDM or co-purified lipids.

Response: *We do not observe ordered extra densities representing detergent molecules or co-purified lipid in the transmembrane region. We have added this observation in line 142-143.*

4. - The presence of blurry densities below the pore in the 2D classes may suggest the existence of artificial tail-to-tail dimers due to solubilization through hydrophobic interactions. The authors should address this issue and consider reconstitution in nanodiscs to stabilize the channel state.

Response: We also noticed blurry densities around the end of the transmembrane domain in 16 out of 37 2D class averages representing MRS2 side views. The blurry densities suggested some MRS2 molecules weakly associate in a tail-to-tail manner. Of note, most of the MRS2 particles exhibit single species in solution, as shown in BN-PAGE (Supplementary Fig. 1c), and no extra density around the transmembrane region was seen in negative-staining EM analysis (Supplementary Fig. 1d). We did extract all particles from 2D class averages indicating the blurry extra density using a larger box size and reclassified in 2D (Fig. R2). Besides the tail-to-tail association, we occasionally also observe tail-to-head and head-to-head associations. Altogether, we speculate these protein associations are likely induced by some local higher particle densities in cryo-EM grids. We do not believe that these associations are relevant in vivo due to the localization of MRS2 in the inner mitochondrial membrane. We thank the reviewer's suggestion of reconstituting MRS2 into lipid environment, which may stabilize the channel and facilitate the elucidation of the still unknown open state. We will undertake it in our future study.

Fig. R2. 2D classification of particles with extra blurry densities. Particles from 16 out of 37 2D class averages of typical side views showing extra blurry densities near the TM domain were re-extracted using a larger box size (1280 px binned to 160 px, ~531 Å) and reclassified in 2D. Various particle-to-particle contacts can be observed, likely due to high particle density on the cryo-EM grid.

5. - The authors should provide an explanation or comment on the higher molecular weight bands observed in BN-PAGE.

Response: We observed that a protein contaminant (mitochondrial complex III, ~500 kDa) was co-purified with MRS2 using affinity chromatography and was not completely separated from MRS2 using size-exclusion chromatography. The peak of the contaminant partially overlapped

with that of MRS2 (Supplementary Fig. 1a), and it can be seen as additional faint bands in the SDS-PAGE gel (Supplementary Fig. 1b). A few 2D class averages of mitochondrial complex III particles can also be seen in our negative-staining EM dataset (Fig. R3). The higher molecular weight bands observed in the BN-PAGE gel likely belongs to the co-purified protein contaminant (mitochondrial complex III, ~500 kDa). We have added this information in the figure legend of Supplementary Fig. 1 highlighted in yellow.

Fig. R3. Negative-staining EM analysis of purified human MRS2 showing mitochondrial complex III as protein contaminant. Class averages 1, 3, 20, 37 and 44 represent mitochondrial complex III in the sample.

6. - The identification of two additional Mg^{2+} ions along the pore is surprising and should be discussed in more detail.

Response: In our MRS2- Mg^{2+} structure, we identified four Mg^{2+} ions in the center of the pore. Compared to the recently published MRS2 structures (Li et al., Nat. Comms. 2023, 14(1), 4713), two additional Mg^{2+} (Mg^{2+-2} , 3) have been identified in our structures (both MRS2- Mg^{2+} and MRS2-EDTA). The presence of the two additional Mg^{2+} were confirmed in the non-symmetric CI map, eliminating the possibility that they are artifacts generated by symmetry refinement (Supplementary Fig. 4b). Mg^{2+-2} and Mg^{2+-3} are coordinated by a ring of hydroxyl oxygens of T346 and carbonyl oxygens of N339 with a distance of 5.2 Å and 4.3 Å, respectively, suggesting

the Mg²⁺ ions are in hydrated form. The additional Mg²⁺ identified in our structure can be reasoned by the higher Mg²⁺ concentration (40 mM) in our sample, compared to 20 mM used in the previous publication (Li et al., Nat. Comms. 2023, 14(1), 4713). For our MRS2-EDTA structure, the MRS2 sample was prepared in the presence of 40 mM Mg²⁺ prior to dialysis and addition of EDTA. The presence of Mg²⁺ in the pore of MRS2-EDTA is likely trapped by the high affinity GMN motif and the gating R332/M336-rings.

More discussion about the two additional Mg²⁺ is now included in line 324-330.

Reviewer #3 (Remarks to the Author):

The manuscript “Cryo-EM structures of human magnesium channel MRS2 reveal gating and regulatory mechanisms” by Lai and coworkers describes substantial progress in our understanding of the structural basis and mechanism of the human mitochondrial Mg²⁺ transporter.

The paper presents two high-resolution cryo-EM structures of the transporter obtained under Mg-saturating conditions (inactive form) and in the presence of EDTA, which removes Mg²⁺ from regulatory sites but does not empty the pore entirely. This indicates high-affinity pore-ion interactions and also suggests that the ions may play a structure-stabilizing role. The resolution allowed the authors to conclude which Mg²⁺ ion is directly coordinated by the sidechain or backbone oxygens and which one is coordinated through water (Mg²⁺-1, for instance). The identification of R332 and M336 (main constriction) as two partially redundant gates (both have to be mutated to produce a gain-of-function effect) is a novel and interesting result. The identification and disruption of the R116-E291 salt bridge between adjacent soluble domains, which results in a higher transporter activity, are also important results showing a regulatory mechanism different from bacterial homologs. The discovery of the 71-residue-long mitochondrial transit peptide at the N-terminus of MRS2 is novel too.

1. What makes this work distinct from the recently published paper by Li and coworkers (Ref 45)? The authors of that previous paper found only two Mg²⁺ ions in the pore, and they proposed the role of Cl⁻ ions as co-ions that help Mg²⁺ overcome the ring of R332 residues. They performed MD simulations illustrating the degree of pore hydration (with the accuracy of the force field) and emphasized the role of membrane potential in driving Mg²⁺ inside. But the present paper better shows the effects of mutations, which were tested in-vivo, and the toxicity of Ni²⁺, which also permeates through the transporter, and the new R116-E291 salt bridge with its regulatory role. In the two papers, the structures are essentially identical, which gives credit to both groups and shows how reliably the same structure was determined in different detergents. In other aspects, the two papers perfectly complement each other and both deserve attention.

Response: We thank the reviewer for the appreciation, and pinpointing the novelty of this work.

2. About Mg²⁺-free condition (1 mM EDTA), under which three Mg²⁺ ions remain in the pore. For Mg²⁺ - EDTA in HEPES, K_d is about 1 μM (O'Brien, J. Chem. Educ. 2015, 92, 1547–1551), which is indeed comparable to the GMN-Mg²⁺ affinity. I would call these conditions not Mg-free, but rather low-Mg condition. That is why the ion remains in the site. If you try a higher EDTA concentration or a stronger chelator, in the absence of Mg²⁺, the GMN motifs and other domains may get disordered. The fact that GMN binds Mg²⁺ so tightly may have a clear functional consequence, namely, a very slow Mg²⁺ transport rate. Because the GMN site is strictly 'in-series' with the rest of the pore pathway, the Mg²⁺ unbinding kinetics might be limiting unless GMN loops come apart and release the ion in the active state. The result begs for experimental measurements of Mg²⁺ unbinding rate. This tight binding is in stark contrast to K⁺ channels where the ion entering the selectivity filter from water experiences zero free energy change (Varma and Rempe, BJ 2007, 93:1093).

Response: We agree that the condition is not Mg²⁺-free upon the addition of 1 mM EDTA. We have revised the “Mg²⁺-free condition” to “EDTA condition” and mentioned the addition of EDTA removed “most of the Mg²⁺” instead of “all Mg²⁺” in solution. We are also thankful for the reviewer’s elaboration on the implication of Mg²⁺ binding by the GMN motif. As demonstrated previously (Dalmás et al., PNAS 2014, 111(8), 3002-3007), the GMN motif possesses high affinity towards Mg²⁺ with an estimated K_D of 1.3 μM. The tightly bound Mg²⁺ by the GMN motif blocks the entry of other cations into the permeation pathway, contributing to ion selectivity to the CorA channel. The investigation of Mg²⁺ binding and unbinding kinetics of MRS2 would be useful to understand the ion transport mechanism and ion selectivity of this channel in future.

The work is implemented according to high standards with a lot of details in the supplement, so I have no technical comments.

3. I have only one request for a short additional experiment. The mitochondrial MRS2 protein was expressed in Expi293F cells (apparently derived from the human HEK293 cell line). My question is whether the expressed protein is localized strictly in mitochondria or elsewhere. The standard immunofluorescence with anti-Flag antibodies may quickly give an answer.

Response: As suggested by the reviewer, we performed subcellular localization analysis of MRS2 in Expi293F cells. Expi293F cells expressing MRS2-GFP with or without its mitochondrial transit peptide (MTP) were imaged 1 day after transfection. Cells were stained with MitoTracker Red or

ER-Tracker Red prior to imaging. This result is now included in Supplementary Fig. 1f-g and described in line 123-127 highlighted in yellow. It shows that full-length MRS2-GFP localizes in mitochondria (Supplementary Fig. 1f), while MRS2 without MTP (MRS2(71-443)-GFP) can no longer be imported into mitochondria and largely localizes in the ER (Supplementary Fig. 1g).

4. Also, is overexpression of the R332A/M336A mutant toxic for the cells?

Response: *Although the R332A/M336A mutations significantly enhance MRS2 channel activity, we do not observe toxicity of overexpressing R332A/M336A in both Mg²⁺-auxotrophic E. coli and BL21-(DE3). The growth (in term of colony forming unit and size) of E. coli expressing the R332A or M336A single mutants and double mutant is almost the same to the E. coli with empty vector, as shown on the control agar plates supplemented with 0.1 mM IPTG (Fig. 2f-g).*

Minor points and suggested edits:

5. Line 86: ‘challenging’ can be replaced by ‘limited’

Line 112: I would replace ‘using SDS-PAGE’ with ‘based on SDS-PAGE’

Line 169: ‘carboxylic acids’ better to replace with ‘carboxylates’ or ‘carboxylic oxygens’

Line 178: ‘capable of interrupting the ion flow’ instead of ‘regulating Mg²⁺ translocation’

Response: *We thank the reviewer for these suggestions. We have revised the manuscript accordingly.*

6. Lines 216-223: This paragraph describes a very important difference in ion coordination, either directly by sidechain or backbone oxygens or through intercalating water. I would suggest putting a special figure illustrating these types of coordination and two or three phrases in the Discussion. Coordination through water is characteristic specifically for Mg²⁺ and might be one of the components of the selectivity mechanism.

Response: *We are thankful for the suggestions from the reviewer. We observed multiple water molecules around the Mg²⁺-5 and Mg²⁺-6, with some water molecules in coordination of Mg²⁺ with distances of 2.1 - 3.1 Å. We have updated Fig. 3b showing water molecules around Mg²⁺ and included a description of these water molecules in line 223-224 highlighted in yellow.*

7. Line 278: I would add ‘Besides naturally present high membrane potential across the inner mitochondrial membrane, a negative surface potential at the pore-facing side....’

Line 306: I would add: ‘... which disrupts a salt bridge between adjacent subunits and mimics low-Mg²⁺ conditions, shows a gain of function...’

Response: We thank the reviewer for the suggestions. We have revised the manuscript accordingly.

8. One question about Fig. 3d. It needs to be better discussed whether Mg²⁺-6 when present in the second regulatory site stabilizes or disrupts the R116-E291 salt bridge. With the E291K mutation, the channel becomes more active, and therefore Mg²⁺-6 may stabilize this bridge.

Response: The side chains R116 and E291 is far from Mg²⁺-6 (with a distance ~12 Å). We anticipate the influence from Mg²⁺-6 to R116-E291 is not profound.

9. Please increase the fonts in the Supplemental figures, some panels are not legible.

Response: We have increased the font size in the Supplementary figures accordingly. All the font size is now at least equal to or larger than 5pt, as suggested by the journal guideline.